# Assessment of Electrical Brain Activity of Healthy Volunteers Exposed to 3.5 GHz of 5G Signals within Environmental Levels: A Controlled–Randomised Study

**DOI:** 10.3390/ijerph20186793

**Published:** 2023-09-21

**Authors:** Layla Jamal, Lydia Yahia-Cherif, Laurent Hugueville, Paul Mazet, Philippe Lévêque, Brahim Selmaoui

**Affiliations:** 1Department of Experimental Toxicology and Modeling (TEAM), Institut National de l’Environnement Industriel et des Risques (INERIS), Parc Technologique Alata, BP 2, 60550 Verneuil-en-Halatte, France; layla.jamal@ineris.fr; 2PériTox Laboratory (UMR_I 01), INERIS/UPJV, INERIS, 60550 Verneuil en Halatte, France; 3Paris Brain Institute (ICM), Center for NeuroImaging Research (CENIR), Sorbonne University, INSERM U1127, CNRS UMR7225, Pitié-Salpêtrière Hospital, 75013 Paris, France; lydia.yahia-cherif@icm-institute.org (L.Y.-C.); laurent.hugueville@icm-institute.org (L.H.); 4Technical Centre for Mechanical Industries (CETIM), 52 Avenue Félix Louat, 60300 Senlis, France; paul.mazet@cetim.fr; 5XLIM Research Institute, University of Limoges, UMR CNRS 7252, 123 Avenue Albert Thomas, 87000 Limoges, France; philippe.leveque@unilim.fr

**Keywords:** 5G, radio frequencies, humans, EEG, resting wake state, electrical brain activity

## Abstract

Following the recent deployment of fifth-generation (5G) radio frequencies, several questions about their health impacts have been raised. Due to the lack of experimental research on this subject, the current study aimed to investigate the bio-physiological effects of a generated 3.5 GHz frequency. For this purpose, the wake electroencephalograms (EEG) of 34 healthy volunteers were explored during two “real” and “sham” exposure sessions. The electromagnetic fields were antenna-emitted in an electrically shielded room and had an electrical field root-mean-square intensity of 2 V/m, corresponding to the current outdoor exposure levels. The sessions were a maximum of one week apart, and both contained an exposure period of approximately 26 min and were followed by a post-exposure period of 17 min. The power spectral densities (PSDs) of the beta, alpha, theta, and delta bands were then computed and corrected based on an EEG baseline period. This was acquired for 17 min before the subsequent phases were recorded under two separate conditions: eyes open (EO) and eyes closed (EC). A statistical analysis showed an overall non-significant change in the studied brain waves, except for a few electrodes in the alpha, theta, and delta spectra. This change was translated into an increase or decrease in the PSDs, in response to the EO and EC conditions. In conclusion, this studhy showed that 3.5 GHz exposure, within the regulatory levels and exposure parameters used in this protocol, did not affect brain activity in healthy young adults. Moreover, to our knowledge, this was the first laboratory-controlled human EEG study on 5G effects. It attempted to address society’s current concern about the impact of 5G exposure on human health at environmental levels.

## 1. Introduction

Radio frequencies (RF) are electromagnetic signals, ranging from 10 MHz to 300 GHz, and are used in wireless communication, among other technologies. These radiations have been classified as possibly carcinogenic to humans [1] due to the limited and contradictory evidence found in animal and human research. With the advent of mobile phones, given their proximity to the head during a phone call, the brain is one of the most exposed organs to this technology. Consequently, researchers have increasingly focused on studying cerebral activity to investigate the potential health effects resulting from this exposure.

Electrical brain activity generates five main wavebands associated with different physiological functions [2]. “Delta” waves are the slowest and oscillate between 0.5 and 4 Hz. They are mostly observed during deep sleep and are believed to be involved in memory consolidation and physical restoration. On the other hand, “theta” waves, which are implicated in memory formation and creativity, range from 4 to 8 Hz. They are mostly prominent during light sleep, daydreaming, and drowsiness. In the awake relaxed state, “alpha” bands are more likely to be seen in the parietal and occipital lobes, where they fluctuate between 8 and 12 Hz, when the eyes are closed. Nevertheless, for more focused and decision-making activities, “beta” bands oscillating in the range of 12–35 Hz are more involved, particularly in the frontal and parietal lobes. However, the fastest “gamma” waves, with frequencies exceeding 35 Hz, are more frequently experienced in high cognitive processing. They are consolidated in all brain areas, and their regulation is associated with effective learning abilities [3]. However, it is noteworthy that these wavebands are not mutually exclusive and can overlap with one another depending on the person’s activity. In addition, their frequency classification varies between studies and is related to age, health status, sex, the eye condition, and many other factors [4]. Many publications have investigated these bands and their responses to several RF signals. However, the most consistent effect discussed in the literature is observed with the alpha rhythm, with increased [5,6,7,8,9,10,11,12,13,14,15,16], decreased [17,18,19,20,21], or unmodified activity [5,8,12,22,23,24,25,26,27,28,29,30,31,32,33].

Nonetheless, with the evolution of wireless communication, fifth-generation RF (5G) has emerged, promising a faster connection and faster data transfers, among other advantages. One of the first deployed bands was 3.4–3.8 GHz. Higher frequencies around 26 GHz will be introduced later for infrastructure and industrial use. Nevertheless, concerns regarding the health impact of 5G networks are still expressed. Thus far, the published studies addressing this issue are limited and probably do not represent the current regulations and safety limits imposed by health authorities [34,35]. To date, the few studies conducted on this topic have been inconclusive [36,37,38,39,40].

Consequently, this project aimed to explore the electrical activity of the brain (beta, alpha, theta, and delta), along with other parameters of the autonomic nervous system, in healthy human participants when exposed to 5G in a restful waking environment. The frequency generated was 3.5 GHz, with exposure levels below the regulatory limits of public use [35] and with an electrical field intensity similar to the one found in the outdoor environment, according to recent measurements carried out in South Korea [41] and other countries [42]. To our knowledge, this is the first study to investigate the effects of 5G signals on brain activity in young healthy volunteers. In this article, we only discuss brain activity outcomes; we will present other parameters at a later time.

## 2. Materials and Methods

### 2.1. Volunteers

Thirty-four healthy volunteers (seventeen males and seventeen females with a mean ± standard deviation [SD] age of 26.6 ± 4.7 years and a mean ± SD body mass index of 23.3 ± 4.1 kg/m^2^) were included under strict inclusion criteria. The participants had regular sleeping habits (sleep–wake cycle of 11 p.m. to 8 a.m. ± 1 h) with no chronic or acute illness or disability. They were non-smokers with no identified electromagnetic hypersensitivity. Pregnant (confirmed by the NADAL hCG pregnancy urinary test, ref: 152002, nal von minden GmbH, Moers, Germany) or nursing women were excluded, along with drug and narcotics users (confirmed by NarcoCheck urinary test, ref: DOA-M10-3B, Kappa City Biotech SAS, Montluçon, France). In addition, for female participants, only those with regular menstrual cycles (25–32 days) and no hormonal contraception intake were selected. Their inclusion sessions were conducted during the follicular phase of their menses.

The experimental protocol (ID-RCB n°:2020-A03127-32) was approved by the French national ethical committee “CPP Sud-Ouest et Outre-Mer 1”, and the selected volunteers were only allowed to participate after signing an informed consent form. Furthermore, the participants were asked to refrain from consuming any stimulating beverages or substances (caffeine, alcohol, chocolate, etc.) 24 h before each experimental session.

It is noteworthy that the recruitment process included participants from diverse cultural and multinational backgrounds. Moreover, sex was determined through a self-report measure based on the type of reproductive system. The volunteer characteristics are detailed in Appendix A.

### 2.2. Study Design and Experimental Protocol

The study was triple-blinded for the volunteers, the experimenter (L.J.), and the data analysts (L.J. and L.Y.C.). All participants underwent two counterbalanced cross-over and randomised “real” and “sham” exposure sessions. B.S. determined the random allocation, whereas L.H. ensured the blinding process.

The time interval between the sessions was no longer than one week. In addition, for each participant, both sessions took place at the same time of the day, either in the morning (9–11 a.m. ± 30 min) or in the afternoon (2–4 p.m. ± 30 min), to prevent any potential modification related to the circadian rhythm. Each session lasted around 2 h (120 ± 30 min) and contained a “baseline” and a “post-exposure” period of 17 min each with no RF exposure. These were separated by either a “real” or “sham” exposure period of 25 min and 30 s. These periods included 2–3 runs of 8 min and 30 s, as shown in Figure 1. Each run started with electrodermal activity (EDA) recording for 2 min and 30 s with the eyes-open (EO) condition. This block was followed by 3 min of another EO condition along with a subsequent 3-min eyes-closed (EC) one. Electroencephalograms (EEGs) and electrocardiograms (ECGs) were continuously acquired in each run. The experimenter used previously recorded vocals to guide the volunteers throughout the sessions. Additionally, to reduce ocular artefacts, a fixation point was displayed on a screen at a distance of 1 m from the participants, on which they could focus. Moreover, the participants were encouraged to maintain modest movements to minimise muscle artefacts.

In addition, temperature measurements along with some salivary biomarkers of stress, namely cortisol, α-amylase, and chromogranin-A, were analysed to explore potential alterations in the autonomic nervous system. However, we only discuss the EEG outcomes here; the other parameters will be presented in the future.

### 2.3. Exposure System

Volunteers were exposed at a mean ± SD ambient temperature of 25.4 ± 0.2 °C in an electrically shielded room. They received RF emissions via a horn antenna (Schwarzbeck Mess-Elektronik oHG, Schönau, Germany) placed 1.2 m away and 45 degrees to the right of each participant. During the sessions, a dosimeter (MVG/EME Spy Evolution, MVG INDUSTRIES|SATIMO, Plouzané, France) was installed in the data acquisition room to verify the exposure parameters. However, the 5G signal generator (SMB100A [1406.6000.02], Rohde & Schwarz GmbH & Co KG, Munich, Germany) wired to a 25-W power amplifier (SX 40/15, Prâna, Brive-la-Gaillarde, France) was placed in a separate room.

Figure 2 illustrates the setup for far-field exposures, which were pulse-modulated (577 µs/4.6 ms) with an electrical field root-mean-square (RMS) intensity of 16 V/m and 12 V/m at the head and trunk levels, respectively. Due to modulation, the mean electric field RMS intensities became 2 and 1.5 V/m, respectively. This was verified with field meter measurements (NBM 550, Narda Safety Test Solutions GmbH, Pfullingen, Germany). Furthermore, the estimated peak power density (PD) was 0.68 W/m^2^, obtained with the following formula: PD = Pt × G/(4 π r^2^), where Pt is the antenna’s power equal to 0.42 W, G is the antenna’s power gain equal to 14.5 (11.6 dB), and r represents the distance from the antenna corresponding to 1.2 m. This PD value is consistent with the electrical field intensity of 16 V/m.

RF numerical dosimetry, i.e., quantifying the energy absorbed by tissues exposed to an electromagnetic field, is fundamental to ensure the reproducibility and reliability of bioelectromagnetic results. The specific absorption rate (SAR) in W/kg is the electromagnetic power dissipated per unit mass and is the metric used to define RF exposure and basic restrictions. Therefore, numerical simulations were performed to determine the exposure SAR at 3.5 GHz. A custom finite difference time domain (FDTD)-based code was used to extract SAR levls averaged over the head and brain tissues [43,44,45]. The human head model was based on a computer-segmented data set created by a collaboration between the National University of Singapore and Johns Hopkins University. The original images were based on photographic data from the Visible Human Project [46] created by the National Library of Medicine and the University of Colorado Health Sciences Center. Our study used an initial anatomical data set with a 1 mm^3^ voxel size. At 3.5 GHz, the dielectric permittivity, conductivity, and density of each tissue were extracted from the IT’IS Foundation database [47] by relying on Gabriel dispersion relationships [48,49]. Due to the distance between the human head and the horn antenna, the wave emitted by the antenna can be approximated by a plane wave with a vertical electric field polarisation and a 2 V/m intensity, corresponding to a mean PD of 11 mW/m^2^.

Head-averaged SAR (HASAR) and brain-averaged SAR (BASAR) were assessed to characterise the current exposure. The human model used had a head mass of 5.5 kg and a brain mass of 1.3 kg. At 3.5 GHz, the HASAR and BASAR were 0.037 ± 0.11 mW/kg and 0.008 ± 0.019 mW/kg, respectively.

### 2.4. Signal Acquisition and Data Processing

The EEGs were recorded in a dimly lit and electrically shielded room with a wakeful resting state, using a 64-channel cap (actiCAP-Snap, EASYCAP GmbH, Brain Products GmbH, Wörthsee, Germany) with active electrodes (actiCAP-slim electrodes, EASYCAP GmbH, Brain Products GmbH, Wörthsee, Germany). The 64 electrodes (Fp1, Fp2, AF7, AF3, AFz, AF4, AF8, F7, F5, F3, F1, Fz, F2, F4, F6, F8, FT9, FT7, FC5, FC3, FC1, FC2, FC4, FC6, FT8, FT10, T7, C5, C3, C1, Cz, C2, C4, C6, T8, TP9, TP7, CP5, CP3, CP1, CPz, CP2, CP4, CP6, TP8, TP10, P7, P5, P3, P1, Pz, P2, P4, P6, P8, PO7, PO3, Poz, PO4, PO8, O1, Oz, O2, and Iz) were arranged according to the extended 10–20 (10%) system [50] with an additional ground (Gnd) and reference (FCz) electrode. Moreover, a disposable ground electrode (Ambu^®^ Neuroline 720, Neurology auto-adhesive surface electrodes, Ambu Sarl, Bordeaux, France) was placed on each participant’s left scapula.

Six disposable electrodes (Ambu® Neuroline 720, Neurology auto-adhesive surface electrodes, Ambu Sarl, Bordeaux, France) were used to detect blinks and cardiac artefacts. Two electrodes were placed near the outer canthus of each eye for the horizontal electrooculogram (EOG), while two more were positioned above and below the right eye for the vertical EOG. In contrast, one electrode was placed on the right clavicle and another on the left lower hip for ECG monitoring.

A sampling frequency of 1000 Hz and a bandwidth of 0.016–250 Hz were used for data acquisition by the Brain Vision recorder software (version 1.23.0003). With the help of a conductivity gel (Electro Gel, Electro Cap Center B.V., Nieuwkoop, the Netherlands), the electrical impedance was maintained below 30 kΩ.

Regarding signal pre-processing, with the aid of MNE-python (version 0.19.2) [51], a bandpass filter of 1–40 Hz was applied along with an artefact repair algorithm based on the method of fast independent component analysis (FastICA) [52]. Subsequently, whitened data underwent a fast Fourier transformation: Welch’s method with the Hamming window function of 4096 points per window and 50% of overlap between segments to first create separate EO and EC epochs and then compute the power spectral density (PSD) of the beta (12–35 Hz), alpha (8–12 Hz), theta (4–8 Hz), and delta (1–4 Hz) brain waves for each recorded run. Gamma waves were not computed because the current protocol contained no cognitive tasks requiring attention or high-performance abilities.

Furthermore, log-transformed PSD values were used for the final data analysis to ensure that the data followed a normal distribution. Subsequently, these transformed values were averaged based on their respective time periods, which included baseline measurements (run 01 and run 02), exposure periods (run 03, run 04, and run 05), and post-exposure periods (run 06 and run 07). This was achieved by using an adapted MNE-python (version 0.19.2) script code. Following this averaging, the statistical analysis was carried out. Additionally, baseline-corrected exposure and post-exposure data were computed to conduct a different type of statistical analysis, as elaborated in the subsequent section. 

### 2.5. Statistical Analysis

The sample size was calculated with the G*power software (version 3.1.9.2), considering a statistical inference of 80% power with a medium effect size (0.5) and 95% confidence interval for analysis of variance (ANOVA) F-tests.

Following data processing and curation, statistical tests were conducted using a customised MATLAB toolbox (version 2019b) [53]. A three-way repeated-measures ANOVA was used to examine the effects of the exposure condition (sham exposure vs. real exposure), time period (baseline vs. exposure or baseline vs. post-exposure), and eye condition (EC vs. EO), as well as their interactions, on log-power values within each frequency band (beta, alpha, theta, and delta). Additionally, a one-way repeated-measures ANOVA was performed on the baseline-corrected exposure and post-exposure log-power values for each frequency band, comparing the two exposure conditions separately for EC and EO recordings. To control the conducted multiple ANOVAs (in multiple sensors), a cluster-based permutation test was performed for each frequency band (not presented in this article). This method was implemented in Fieldtrip (Toolbox in MATLAB 2019b) for *t*-tests [54], and it was extended to the repeated-measures ANOVA. Statistical significance was defined as *p* < 0.05.

## 3. Results

As mentioned in the statistical plan, we conducted a three-way repeated-measures ANOVA—5G exposure (Factor 1: “real” vs. “sham” conditions), time period (Factor 2: baseline vs. exposure or post-exposure phases), and eye condition (Factor 3: EO vs. closed EC)—on each brain wave frequency band. We present the findings of each assessed factor, including those of 5G. However, for simplicity, only the electrodes showing significant outcomes related to 5G (Factor 1) are indicated with an arrow in the figures.

### 3.1. Beta Spectral Power

#### 3.1.1. 5G Exposure (Factor 1)

The three-way ANOVA showed no significant difference between “real” and “sham” sessions (Factor 1) in the overall studied electrodes, except for Fp1 (*p* = 0.0194) and TP7 (*p* = 0.0460) in the exposure period and only for Fp1 (*p* = 0.0052) in the post-exposure period when compared with those at baseline (Figure A1). However, following the one-way ANOVA for the baseline-corrected data, the significance between the “real” and “sham” sessions did not persist in the previously mentioned electrodes in both periods, as shown in Figure 3. This was also confirmed by the cluster-based permutation test.

#### 3.1.2. Time Period (Factor 2)

Moreover, regarding the effect of the time period, there was no significant difference between the baseline and the exposure periods following three-way ANOVA. Likewise, when we compared the baseline and post-exposure periods, only the AFz (*p* = 0.0138), Fz (*p* = 0.0196), and C2 (*p* = 0.0054) electrodes showed a significant difference (Figure A1).

#### 3.1.3. Eye Condition (Factor 3)

Regarding the eye condition, the difference was notably significant in 40 out of 64 electrodes when comparing the baseline and exposure periods (Fp1, Fp2, AF7, AF3, AFz, AF4, AF8, F7, F5, F3, F1, F2, F4, F6, F8, FT9, FT7, FC5, FC3, FC2, FC4, FC6, FT8, FT10, T7, C5, T8, Pz, P2, P4, P6, PO7, PO3, Poz, PO4, PO8, O1, Oz, O2, and Iz). Moreover, 39 out of 64 electrodes were significant when comparing the baseline and post-exposure periods (Fp1, Fp2, AF7, AF3, AFz, AF4, AF8, F7, F5, F3, F2, F4, F6, F8, FT9, FT7, FC5, FC3, FC4, FC6, FT8, T7, C5, C6, T8, Pz, P2, P4, P6, P8, PO7, PO3, Poz, PO4, PO8, O1, Oz, O2, and Iz), as shown in Figure A1.

#### 3.1.4. Interaction among Factors

Regarding the interaction of 5G exposure with other factors, only T7 (*p* = 0.0334) showed a significant result between 5G and the eye condition (Factor 1 × Factor 3) when we compared exposure and baseline data. On the other hand, FC6 (*p* = 0.0454), FT8 (*p* = 0.0417), and FT10 (*p* = 0.0317) displayed a similar outcome between 5G and the eye condition when we compared baseline and post-exposure data. Nevertheless, there was no interaction between 5G exposure and the time period (Factor 1 × Factor 2). Regarding the interaction among the three studied factors combined (Factor 1 × Factor 2 × Factor 3), there was no significance except for FT8 (*p* = 0.0169), FT10 (*p* = 0.0323), and T7 (*p* = 0.0072) when we compared the baseline and post-exposure periods. All of the above details are illustrated in Figure A1.

### 3.2. Alpha Spectral Power

#### 3.2.1. 5G Exposure (Factor 1)

There was no significant change in the studied electrodes due to the 5G factor following the three-way ANOVA, except for the AF7 electrode (*p* = 0.0452), when we compared the baseline and post-exposure periods of both sessions (Figure A2).

Following the one-way ANOVA of the baseline-corrected data, the PSD values of the overall electrodes were not significantly different between “real” and “sham” exposures, except for P7 (*p* = 0.0449) and TP10 (*p* = 0.0265), in the EC condition of the exposure period. These displayed a slight power decrease during the genuine exposure. However, in the post-exposure period, only F5 showed a significant decrease (*p* = 0.0321) in the EC condition due to the 5G effect (Figure 4). All the above-mentioned significant electrodes did not survive the cluster-based permutation test.

#### 3.2.2. Time Period (Factor 2)

Concerning the time period, the comparison of the baseline and exposure periods showed 51 significant electrodes following the three-way ANOVA (F3, F1, F2, F4, F6, F8, FT9, FT7, FC3, FC1, FC4, FC6, FT8, FT10, T7, C5, C3, C1, C2, C4, C6, T8, TP9, TP7, CP5, CP3, CP1, CPz, CP2, CP4, CP6, TP8, TP10, P7, P5, P1, Pz, P2, P4, P6, P8, PO7, PO3, Poz, PO4, PO8, O1, Oz, O2, and Iz). On the other hand, 52 electrodes exhibited a significant difference when we compared the baseline and post-exposure periods (F3, F1, F6, F8, F7, FT9, FT7, FC5, FC3, FC1, FC4, FC6, FT8, FT10, T7, C5, C3, C1, Cz, C2, C4, C6, T8, TP9, TP7, CP5, CP3, CP1, CPz, CP2, CP4, CP6, TP8, TP10, P7, P5, P3, P1, Pz, P2, P4, P6, P8, PO7, PO3, Poz, PO4, PO8, O1, Oz, O2, and Iz), as illustrated in Figure A2.

#### 3.2.3. Eye Condition (Factor 3)

The eye condition factor displayed a significant effect in all the studied electrodes in the exposure and post-exposure periods when compared with the baseline period (Figure A2).

#### 3.2.4. Interaction among Factors

There were no significant interactions among the studied factors aside from a few electrodes in specific factor combinations (Figure A2). Fp2 (*p* = 0.0199), AF8 (*p* = 0.0498), and Oz (*p* = 0.0148) showed a significant interaction between 5G exposure and the eye condition (Factor 1 × Factor 3) in the three-way ANOVA comparison of the baseline and exposure periods. Moreover, the interaction between the time period and the eye condition (Factor 2 × Factor 3) was only significant in AF8 (*p* = 0.0289) and F2 (*p* = 0.0267) when comparing the baseline and exposure periods, and in T7 (*p* = 0.0383) and TP9 (*p* = 0.0455) when comparing the baseline and post-exposure periods.

### 3.3. Theta Spectral Power

#### 3.3.1. 5G Exposure (Factor 1)

Based on the three-way ANOVA results, the 5G factor did not modify the PSD values of theta waves (Figure A3).

Regarding the baseline-corrected data and the 5G factor, the one-way ANOVA revealed no significant difference in the overall analysed electrodes except for the EC condition in P2 (*p* = 0.0401) during exposure, along with P2 (*p* = 0.0181), P6 (*p* = 0.0373), PO4 (*p* = 0.0301), PO8 (*p* = 0.0340), and O2 (*p* = 0.0490) in the post-exposure period. The PSD values of the above-mentioned electrodes increased during and after “real” 5G exposure (Figure 5). However, statistical correction performed by the cluster-based permutation test did not reveal any significance in the analysed electrodes.

#### 3.3.2. Time Period (Factor 2)

Following three-way ANOVA, the only significant differences were for P8 (*p* = 0.0475) when comparing the baseline and exposure periods, and Fp1 (*p* = 0.0067), AF3 (*p* = 0.0082), AFz (*p* = 0.0016), AF4 (*p* = 0.0006), AF8 (*p* = 0.0038), F5 (*p* = 0.0470), F3 (*p* = 0.0326), F1 (*p* = 0.0479), Fz (*p* = 0.0092), and F2 (*p* = 0.0297) when comparing the baseline and post-exposure periods (Figure A3).

#### 3.3.3. Eye Condition (Factor 3)

The results of the eye condition variable of the three-way ANOVA showed that 57 out of 64 electrodes were significantly different in the exposure and post-exposure periods compared with the baseline (F7, F5, F3, F1, Fz, F2, F4, F6, F8, FT9, FT7, FC5, FC3, FC1, FC2, FC4, FC6, FT8, FT10, T7, C5, C3, C1, Cz, C2, C4, C6, T8, TP9, TP7, CP5, CP3, CP1, CPz, CP2, CP4, CP6, TP8, TP10, P7, P5, P3, P1, Pz, P2, P4, P6, P8, PO7, PO3, Poz, PO4, PO8, O1, Oz, O2, Iz), as shown in Figure A3.

#### 3.3.4. Interaction among Factors

Only some random electrodes showed significant interactions among the studied factors (Figure A3). The interaction between 5G exposure and the time period (Factor 1 × Factor 2) was significant in P2 (*p* = 0.0232), P6 (*p* = 0.0275), PO4 (*p* = 0.0125), and PO8 (*p* = 0.0122) when comparing the baseline and post-exposure periods. However, for the time period and eye condition interaction (Factor 2 × Factor 3), only T8 (*p* = 0.0371) showed significance when comparing the baseline and post-exposure periods, alongside AF3 (*p* = 0.0287), AFz (*p* = 0.0414), AF8 (*p* = 0.0189), F1 (*p* = 0.0482), F2 (*p* = 0.0249), and F8 (*p* = 0.0493) when comparing the baseline and exposure periods. As for the interaction between the three factors, only the F2 (*p* = 0.0336) electrode displayed a significant result.

### 3.4. Delta Spectral Power

#### 3.4.1. 5G Exposure (Factor 1)

Based on the three-way ANOVA, all electrodes showed a non-significant difference in delta PSDs under “real” and “sham” exposure conditions in the exposure and post-exposure periods (Figure A4). However, in contrast to the overall electrodes, following the one-way ANOVA, only FC5 (*p* = 0.0211) and C3 (*p* = 0.0257) of the baseline-corrected exposure data showed a significant increase in the “real” session in the EC condition compared with the “sham” one (Figure 6). On the other hand, the delta waves of the baseline-corrected post-exposure period displayed a significant increase only in the EO condition in FT9 (*p* = 0.0295), P8 (*p* = 0.0394), and PO8 (*p* = 0.0362), as presented in Figure 6. This significance did not survive the permutation test correction.

#### 3.4.2. Time Period (Factor 2)

We found that 58 of 64 electrodes showed significant differences due to the time period in the three-way ANOVA when we compared the baseline and post-exposure periods (Fp1, Fp2, AF7, AF3, AFz, AF4, AF8, F7, F5, F3, F1, Fz, F2, F4, F6, F8, FT9, FT7, FC5, FC3, FC1, FC2, FC4, FC6, FT8, T7, C5, C3, C1, Cz, C2, C4, C6, T8, TP7, CP5, CP3, CP1, CPz, CP2, CP4, CP6, TP8, P7, P5, P1, Pz, P2, P4, P6, PO7, PO3, Poz, PO4, PO8, Oz, O2, and Iz). However, there were no significant differences when we compared the baseline and exposure periods (Figure A4).

#### 3.4.3. Eye Condition (Factor 3)

This factor significantly altered the delta brain waves in both exposure (Fp2, FT7, FC5, FC3, FC1, FC2, FC4, FC6, FT8, C5, C3, C1, C2, C4, C6, T8, TP9, TP7, CP5, CP3, CP1, CP4, CP6, TP8, TP10, P7, P5, P3, P1, Pz, P2, P4, P6, P8, PO7, PO3, PO4, PO8, O1, O2, and Iz) and post-exposure (Fp1, FC4, FC6, FT8, FT7, FC3, C5, C3, C1, C4, C6, TP7, CP5, CP3, CP1, CPz, CP4, CP6, TP8, TP10, P7, P5, P3, P1, Pz, P2, P4, P6, P8, PO7, PO3, Poz, PO4, PO8, O1, and O2) periods when compared with baseline, as illustrated in Figure A4.

#### 3.4.4. Interaction with 5G

As shown in Figure A4, there were no significant interactions in the overall tested combinations following three-way ANOVA, except for a few random electrodes. P8 (*p* = 0.0433) and PO8 (*p* = 0.0203) showed a significant interaction between 5G exposure and the time period (Factor 1 × Factor 2) when we compared the baseline and post-exposure periods. The latter data sets also demonstrated a significant interaction in the FC5 electrode (*p* = 0.0409) between 5G exposure and the eye condition (Factor 1 × Factor 3), alongside the TP10 (*p* = 0.0353) electrode in the time period and eye condition combination (Factor 2 × Factor 3). However, regarding the interaction among the three assessed factors (Factor 1 × Factor 2 × Factor 3), only FC5 (*p* = 0.0017) and C3 (*p* = 0.0161) were significant when we compared the baseline and exposure periods, as well as Poz (*p* = 0.0492) when we compared the baseline and post-exposure periods. 

## 4. Discussion

In this study, we explored the effects of 3.5 GHz, representing one of the earliest exploited 5G bands, via the EEG of healthy human volunteers. Thirty-four participants took part in two triple-blinded, “real” and “sham” exposure sessions. The latter were randomised and counterbalanced in a wake-rested state, where the brain activity was recorded in the EO and EC conditions. Each session contained a baseline recording period to correct the subsequent exposure and post-exposure acquisition phases. The frequency used was pulse-modulated to simulate far-field antenna exposures within public regulation limits [35] emitted by mobile phone operators. Moreover, the applied electrical field intensity was similar to the one currently found in the outdoor environment [41,42]. We analysed the beta, alpha, theta, and delta EEG waveband PSDs to determine the electrical brain activity changes during and after exposure.

Statistical analysis with repeated-measures ANOVA showed no significant changes in the EEG waves in the subjects exposed to the 5G signal, except for a few electrodes in the alpha and beta oscillations, as mentioned in Section 3.1 and Section 3.2. This detected difference could have been random because it did not persist when we applied one-way ANOVA to the baseline-corrected data. However, different electrodes showed significant modulation in these corrected data due to 5G according to the eye condition and time period (during or after exposure). In the EC state, the alpha bands displayed a significant power decrease in the parietal P7 and TP10 electrodes during 5G signal exposure, along with the frontal F5 electrode in the post-exposure period. However, in the theta range, the parietal P2 electrode showed a significant power increase in the exposure and post-exposure periods. Additionally, the PSD values of other parieto-occipital electrodes (P6, PO4, PO8, and O2) were significantly elevated after exposure. As for the delta waves, only the power of the FC5 and C3 electrodes in the left hemisphere of the fronto-central region became significantly greater during RF exposure. In contrast, there was not a significant 5G effect in the studied EEG bands in the EO condition, aside from the baseline-corrected post-exposure data of delta waves. Only the PSD values of one fronto-central (FT9) and two parieto-occipital (P8 and PO8) electrodes significantly increased in the post-exposure period. Nevertheless, these significant differences did not survive the corresponding statistical correction performed by the cluster-based permutation test.

As far as we know, this project is the first pilot study tackling the effects of low-band 5G frequencies on human brain activity. Hence, comparing our outcomes to other 5G studies is not currently possible. On the other hand, because the overall electrodes demonstrated non-significant modulation in the analysed brain waves, we cannot conclude that 3.5 GHz can change EEG profiles. However, our previous investigation of 2G GSM (The Global System for Mobile Communications) signals (900 MHz) revealed a significant decrease in alpha activity during and after exposure in the EC condition [19]. In a subsequent magnetoencephalographic study, the same RF also showed a decrease during EC exposure in the alpha spectral power, mainly at the frontal and parieto-occipital electrodes [55]. Our 5G findings regarding alpha rhythms generally reflected the same reduced patterns (significant in two electrodes) during and after exposure in the EC condition compared with the “sham” exposure. Regarding the other wavebands, our 5G outcomes, although not significant, showed similar reduced trends in the delta oscillations in the EO condition, along with increased overall activity (only significant in two electrodes) with the EC condition during the baseline-corrected exposure period, as shown in our previous 2G study [56]. However, the 5G exposure non-significantly increased the theta waves compared with the “sham” exposure in the EO condition, in contrast to our 2G findings [56]. Nevertheless, during the EC condition, 5G and 2G RF had a similar increasing trend in the baseline-corrected theta waves in the exposure period compared with the “sham” condition. It is noteworthy that we assessed the effects of far-field antenna-emitted 5G. In contrast, in our 2G studies, we explored mobile phone (MP)-emitted RF, representing a near-field scenario. Therefore, exposure in the far field might have influenced the results in our present study and could explain the discrepancy. Furthermore, the 5G signal intensity emanating from the antenna was approximately 2 V/m, whereas, in our previous studies conducted with 2G, the intensity was approximately 10 V/m at the level of the head. Moreover, we calculated the SAR differently. Given that the previous studies employed a local near-field exposure system via the mobile phone, we calculated the SAR and averaged it on 10 g and 1 g of tissue, which resulted in values of 0.49 and 0.70 W/kg, respectively, with a peak SAR of 0.93 W/kg. On the other hand, we applied a far-field antenna-based exposure system in the current study. Hence, the SAR calculations targeted the entire body, with the average head SAR being 0.037 mW/kg. Additionally, from 2G to 5G, the frequency increases from 900 MHz to 3.5 GHz, and it is well known that the higher the frequency, the lower the penetration into the human body. This could partly explain the effect observed with 900 MHz but not with 3.5 GHz. Indeed, higher penetration allows an important energy transfer that could impact the brain. The authors of some reviews have reported that the effects of electromagnetic fields depend on several factors, such as the frequency, intensity, cell type, and exposure duration [42,57,58].

Similarly to our 5G findings, the authors of other studies investigating higher RF (3G, 4G, and Wi-Fi) effects on healthy waking human EEG have found no significant change in the explored cerebral waves due to exposure [12,28,32,33]. As shown by one of these studies [12], the higher RF of 3G failed to induce an effect on alpha oscillations in all the studied age ranges (teenagers, young adults, and elderly). However, 2G exposure enhanced the EO alpha rhythms only in young adults [10], in contrast to another study [59], in which the researchers found a significant alpha activity increase only in the elderly tested population compared with young adults during 2G exposure. Nevertheless, other researchers who explored the 3G and 4G MP emissions in young adults have observed either reduced [21] or elevated [14] cerebral frequencies in the EO condition. Overall, these contradictory EEG results tackling RF generations are likely a consequence of the heterogenous protocols and study designs (the eye condition; the wake, rest, active, or sleep state; the presence or absence of cognitive tasks; age; sex, etc.), as discussed in several reviews [4,58,60], which complicates the interpretation of the findings.

We only included healthy young volunteers. Introducing all age ranges would provide a better representation of the exposed population to explore the age factor as studied elsewhere [12,27,59]. Furthermore, comparing healthy subjects with patients with epilepsy or other neuropathologies would permit a better understanding of the potential clinical implications for brain activity of this new technology.

In addition to the 5G effect, the EEG signals showed a significant outstanding difference between the EO and EC status in all the studied spectra, confirming the literature findings regarding the eye condition influence on EEG activity even in restful environments [61,62]. On the other hand, alpha rhythms were the most modulated in time compared with the other brainwaves, with significant changes in the overall brain regions starting from the exposure period and extending into the post-exposure intervals. Delta waves, however, were only modulated in the post-exposure period, with significant changes in the overall activity. On the other hand, the theta spectrum revealed significant changes only in the frontal lobe in the post-exposure period. Nevertheless, beta waves did not change in time, except for three random electrodes and only in the post-exposure period. We expected these responses because the participants maintained a restful sitting position with no activity during the sessions. Moreover, these findings are consistent with the literature [4,19,56,60,63,64,65,66], thus validating our experimental protocol.

## 5. Conclusions

Our statistical analysis revealed an overall non-significant difference in beta, alpha, theta, and delta brain oscillations relative to 5G exposure. However, a few electrodes in the baseline-corrected exposure and post-exposure periods exhibited significant modulation corresponding to the eye condition only in the alpha, theta, and delta rhythms, which did not survive the posterior statistical correction. Thus, 3.5 GHz within regulated exposure levels and under the current experimental conditions does not affect EEG activity in humans. However, the potential impacts of chronic 5G exposure remain unknown, necessitating large longitudinal cohort studies to understand the biological responses to this modern technology.

## Figures and Tables

**Figure 1 ijerph-20-06793-f001:**
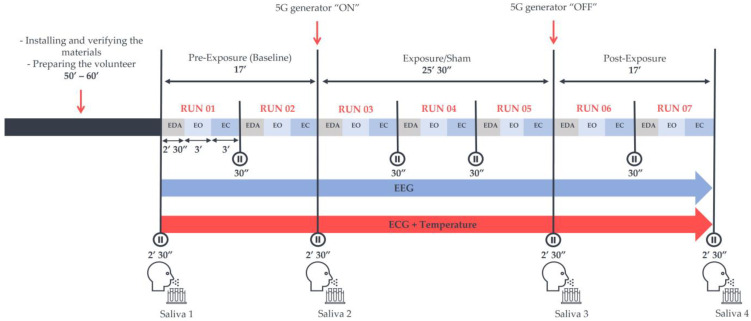
The experimental protocol was preceded by a volunteer preparation phase of approximately 60 min. The protocol included three exposure conditions: the pre-exposure (Baseline), “real” or “sham” exposure, and post-exposure periods. Each exposure and post-exposure period contained 2–3 runs of recording phases that started with the acquisition of electrodermal activity (EDA). Subsequently, 3 min of the eyes-open (EO) condition followed by 3 min of the eyes-closed (EC) condition were recorded in a wakeful resting environment, where an electroencephalogram (EEG) and an electrocardiogram (ECG) were continuously acquired in an electrically shielded room. Four saliva samples were also collected in both sessions to assess some salivary biomarkers of stress.

**Figure 2 ijerph-20-06793-f002:**
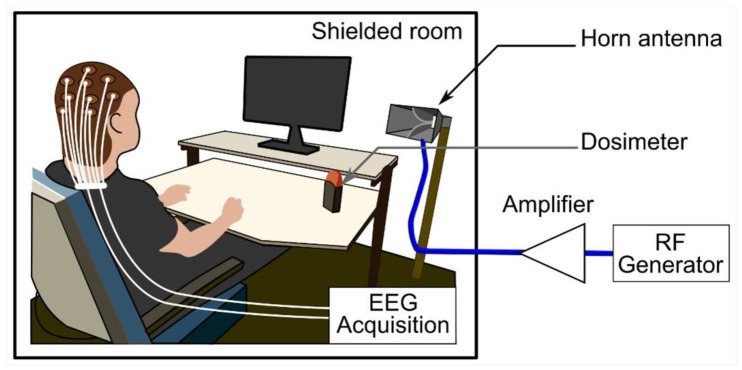
The exposure system setup comprised a horn antenna, a dosimeter to control the exposure parameters, and a 3.5 GHz generator connected to a signal amplifier. The experimental protocol was performed in an electrically shielded room. Abbreviations: EEG, electroencephalography; RF, radiofrequency.

**Figure 3 ijerph-20-06793-f003:**
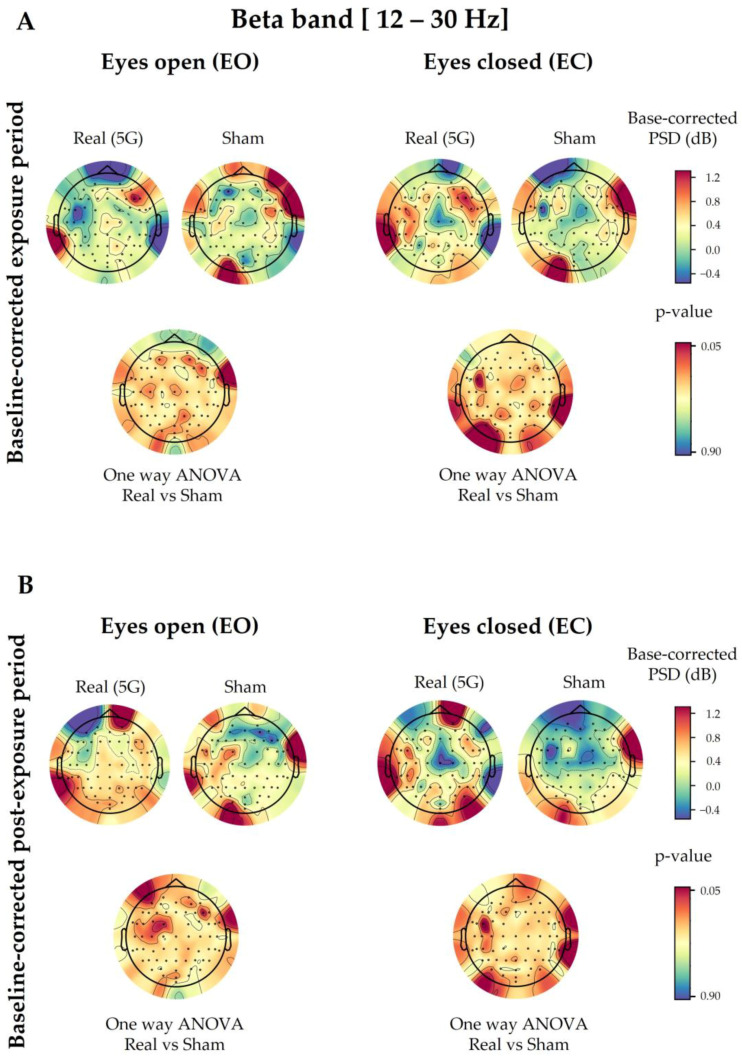
Topographical maps of the beta electroencephalographic (EEG) band of the baseline-corrected exposure (**A**) and post-exposure (**B**) periods. The outcomes are shown separately for the eyes-open (EO) (**left**) and eyes-closed (EC) (**right**) conditions of the “real” and “sham” sessions (upper lines). The upper colour bars on the right indicate the differences in power spectral densities (PSDs) in decibels between the baseline and exposure or post-exposure periods, respectively. The results of one-way analysis of variance (ANOVA) are shown in the lower lines of each corresponding topographical map, with their *p*-value bars displayed on the right (significance level < 0.05).

**Figure 4 ijerph-20-06793-f004:**
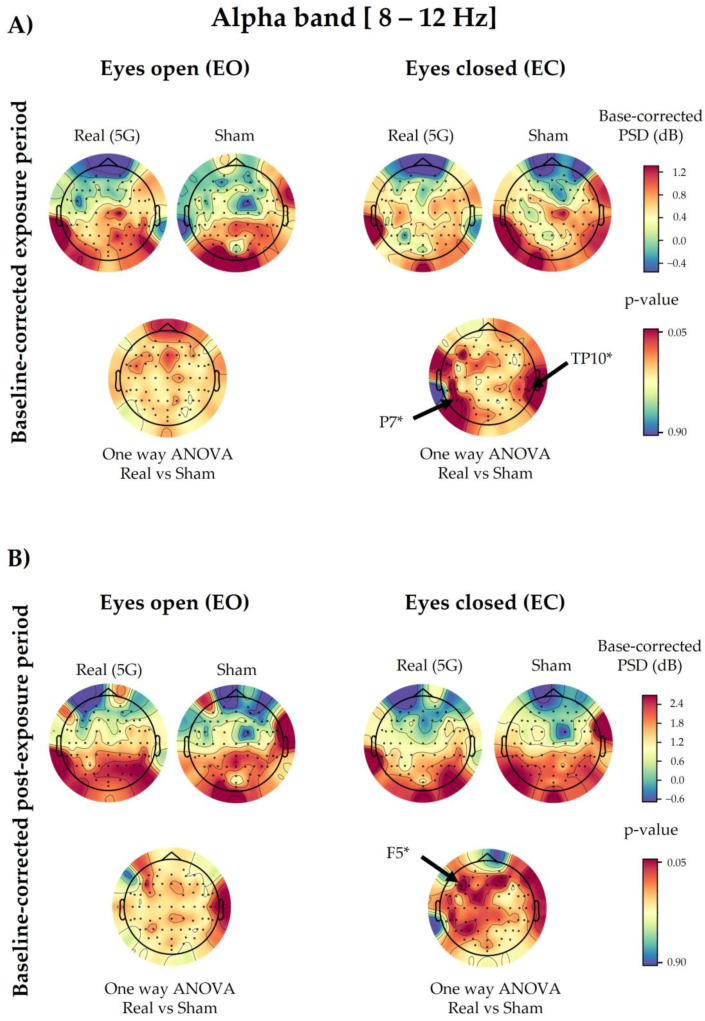
Topographical maps of the alpha electroencephalographic (EEG) band of the baseline-corrected exposure (**A**) and post-exposure (**B**) periods. The outcomes are shown separately for the eyes-open (EO) (**left**) and eyes-closed (EC) (**right**) conditions of the “real” and “sham” sessions (upper lines). The upper colour bars on the right indicate the differences in power spectral densities (PSDs) in decibels between the baseline and exposure or post-exposure periods, respectively. The results of one-way analysis of variance (ANOVA) are shown in the lower lines of each corresponding topographical map. Significant electrodes (*p* < 0.05) are noted with an asterisk (*).

**Figure 5 ijerph-20-06793-f005:**
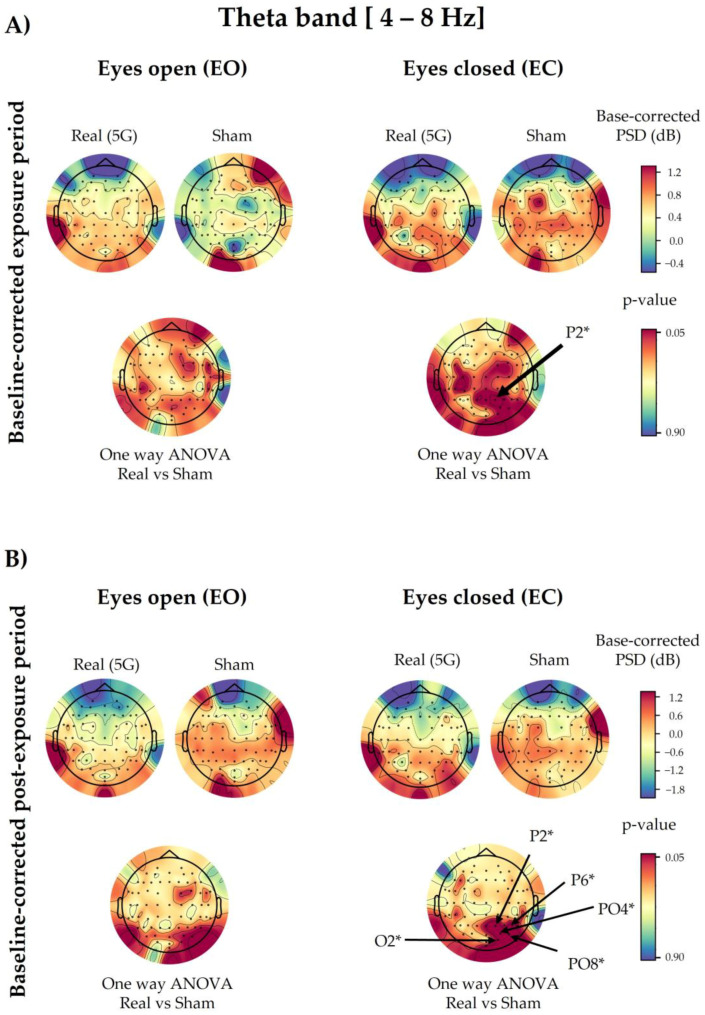
Topographical maps of the theta electroencephalographic (EEG) band of the baseline-corrected exposure (**A**) and post-exposure (**B**) periods. The outcomes are shown separately for the eyes-open (EO) (**left**) and eyes-closed (EC) (**right**) conditions of the “real” and “sham” sessions (upper lines). The upper colour bars on the right indicate the differences in power spectral densities (PSDs) in decibels between the baseline and exposure or post-exposure periods, respectively. The results of one-way analysis of variance (ANOVA) are shown in the lower lines of each corresponding topographical map. Significant electrodes (*p* < 0.05) are noted with an asterisk (*).

**Figure 6 ijerph-20-06793-f006:**
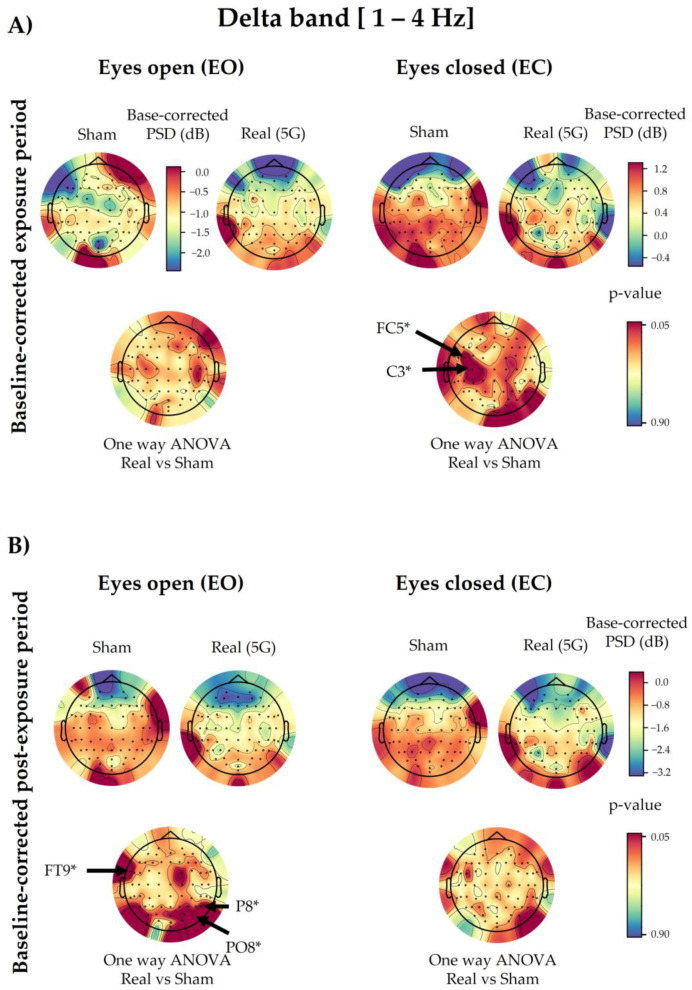
Topographical maps of the delta electroencephalographic (EEG) band of the baseline-corrected exposure (**A**) and post-exposure (**B**) periods. The outcomes are shown separately for the eyes-open (EO) (**left**) and eyes-closed (EC) (**right**) conditions of the “real” and “sham” sessions (upper lines). The upper colour bars on the right indicate the differences in power spectral densities (PSDs) in decibels between the baseline and exposure or post-exposure periods, respectively. Please note that in panel (**A**), only the “sham” session of the EO condition has a different scale bar (on the **right**) for the base-corrected PSD values than the other conditions during the exposure period. The results of one-way analysis of variance (ANOVA) are shown in the lower lines of each corresponding topographical map. Significant electrodes (*p* < 0.05) are noted with an asterisk (*).

## Data Availability

The data presented in this study are available on request from the corresponding author. The data are not publicly available due to confidentiality and privacy considerations.

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
