# Peer review of "Assessment of Electrical Brain Activity of Healthy Volunteers Exposed to 3.5 GHz of 5G Signals within Environmental Levels: A Controlled–Randomised Study"

_ijerph, 2023, doi:10.3390/ijerph20186793_

Round 1
Reviewer 1 Report
As the authors state this is the first human experimental study on effects of 5G on brain activity in the waking resting state. Given that overall there are very few human experimental studies on possible effects of 5G and that there are still open questions regarding RF-EMF effects of 2G, 3G and 4G exposure on brain activity this study is of importance.
I have only some comments:
General comments:
· There seem to be errors in the reference to some Figures in the PDF of the manuscript (lines 110, 145, 243, 262, 293, 296)
· In the text some p-values are displayed with a ‘,’ instead of a ‘.’ – please correct
Minor comments:
· Chapter volunteers: Did all the participants have an alpha resting state EEG or are subjects with other basis EEG rhythms included?
· Chapter Study design and experimental protocol: Were all subjects able to keep their vigilance? Did subjects fall asleep during the 60 min monotonous test session?
· Line 105: The authors mention that sessions took place at the same time of the day. I guess this applies within subjects? Does this also apply across subjects? At what time of the day were sessions scheduled?
· The sentences in lines 122 – 126 replicate the sentences in lines 119 – 122.
· Figure 1 and lines 107 – 115: The authors describe that EEG prior to, during and following exposure was recorded in several runs. It seems that for the analyses results for runs 1 and 2, 3 to 5, as wells as 6 and 7 were pooled.. It is not clear how the values of multiple runs under one condition were included into the analyses. Was there some type of averaging? Was the ANOVA run using all individual values? This is not mentioned in the manuscript and description how this was done is missing.
· Exposure system and signal acquisition: EEG was recorded during RF exposure. Since sensitive electronics like EEG electrode preamplifiers may be prone to RF interference, especially for modulated signals, one could ask, how this was checked for the experimental setup.
· Lines 150 - 154: The estimation is based on an antenna gain of 11.6dB. If this value is taken from the datasheet it applies for far field. Though there is only a slight difference a smaller value should be more appropriate for your distance of 1.2m. As the precision of measurements is limited anyway, e.g. 11dB could be used.
· Lines 231-234: The authors mention that the results of the statistical findings for the others factors …. are displayed in the Supplementary Materials but not discussed. I would prefer to include a discussion of these findings also in the text.
· The Figure A1 is missing in the PDF of the manuscript
· The legend for figures A2, A3 and A4 differ (The one for A1 is missing). In the legend of Figure A2 the description of part B is missing, in the legends for A3 and A4 the description of factors 2 and 3 is missing.
· Figures 6 and 7: If possible, I would prefer to see the results for the Delta band also in one figure (like for the other bands.
Author Response
I would like to thank reviewer n°1 for his/her feedback and comments. Here, you may find in the word document a reply to each addressed point. In the hope our answers (provided in the word document in colour green) and modifications (highlighted in green in the manuscript text) are sufficient.

Reviewer 2 Report
Jamal et al have submitted their ms ” Assessment of electrical brain activity of healthy volunteers ex- posed to 3.5 GHz of the 5G signals within environmental levels: a controlled-randomised study”( ijerph-2557264) for consideration of publication in IJERPH.
The ms describes an experimental study on young and healthy human volunteers that are exposed or sham-exposed to a 3.5 GHz 5G signal in a dedicated shielded room. Experimental endpoints include EEG signals, and also parameters obtained from saliva samples. Results from the latter sampling are not included in this ms but will apparently appear in another publication. The results indicate that after careful statistical analyses, no statistically significant effects on signals representing the alpha, beta, theta, or delta bands are found as consequence of the employed exposure regime.
The study is well designed, performed and described. Only very minor issues are needed to address according to this reviewer.
It is unclear if sessions were one week apart, or as is written on line 105, if the sessions also could be more closely spaced. Also, on line 110 it says that the sham or real exposure periods contained 2 or 3 runs, whereas Fig 1 indicates 3 runs.
A justification of the length of the EDA, eyes open, or eyes closed conditions, respectively would be nice. Are the times based on previous studies?
Lines 122-126 can be deleted since they are duplications of text above.
The SAR-values in this study are very low, seemingly in the microW/kg range. Please provide the calculations that were made to obtain these levels. It would also be relevant to know the SAR-values that the authors used in previous studies employing 2G signals. The E-fields were apparently higher in those studies, but what were the differences regarding SAR?
The writing is in general acceptable, although a careful reading regarding language is recommended before publication. One word that often is misused in many EMF-related papers is ”controversial”, where the more correct alternative would be ”contradictory”, or something similar. I think that this is the case also in this ms. Please check! Furthermore, the statement on line 60 regarding that the most consistent effect on EEG is on the alpha rhythm. Is it possible to elaborate on this? What effects are we talking about? There are a couple of instances where the message ”Error! Reference source not found..” appears in the text. This needs to be cleaned up as well.
Only minor and easily corrected issues were detected. Still, a careful reading of the final ms is recommended.
Author Response
I would like to thank reviewer n°2 for his/her feedback and comments. Here, you may find in the word document a reply to each addressed point. In the hope our answers (provided in colour green in the word document) and modifications (highlighted in yellow in the manuscript text) are sufficient.
